# Allosteric control of an asymmetric transduction in a G protein-coupled receptor heterodimer

Junke Liu[1†], Zongyong Zhang[1†], David Moreno-Delgado[2], James AR Dalton[3,4], Xavier Rovira[2], Ana Trapero[5], Cyril Goudet[2], Amadeu Llebaria[5], Jesús Giraldo[3,4], Qilin Yuan[1], Philippe Rondard[2†], Siluo Huang[1‡*], Jianfeng Liu[1‡*], Jean-Philippe Pin[2‡*]

[1]College of Life Science and Technology, Collaborative Innovation Center for Genetics and Development, and Key Laboratory of Molecular Biophysics of Ministry of Education, Huazhong University of Science and Technology, Wuhan, China; [2]Institut de Génomique Fonctionnelle, CNRS, INSERM, Université de Montpellier, Montpellier, France; [3]Institut de Neurociències and Unitat de Bioestadística, Universitat Autònoma de Barcelona, Barcelona, Spain; [4]Network Biomedical Research Center on Mental Health, Barcelona, Spain; [5]MCS, Laboratory of Medicinal Chemistry and Synthesis, Institute for Advanced Chemistry of Catalonia (IQAC-CSIC), Barcelona, Spain

*For correspondence:
slhuang@mail.hust.edu.cn (SH);
jfliu@mail.hust.edu.cn (JL);
jppin@igf.cnrs.fr (J-PP)

[†]These authors contributed equally to this work
[‡]These authors also contributed equally to this work

Competing interests: The authors declare that no competing interests exist.

**Abstract** GPCRs play critical roles in cell communication. Although GPCRs can form heteromers, their role in signaling remains elusive. Here we used rat metabotropic glutamate (mGlu) receptors as prototypical dimers to study the functional interaction between each subunit. mGluRs can form both constitutive homo- and heterodimers. Whereas both mGlu2 and mGlu4 couple to G proteins, G protein activation is mediated by mGlu4 heptahelical domain (HD) exclusively in mGlu2-4 heterodimers. Such asymmetric transduction results from the action of both the dimeric extracellular domain, and an allosteric activation by the partially-activated non-functional mGlu2 HD. G proteins activation by mGlu2 HD occurs if either the mGlu2 HD is occupied by a positive allosteric modulator or if mGlu4 HD is inhibited by a negative modulator. These data revealed an oriented asymmetry in mGlu heterodimers that can be controlled with allosteric modulators. They provide new insight on the allosteric interaction between subunits in a GPCR dimer.
DOI: https://doi.org/10.7554/eLife.26985.001

## Introduction

Many cell surface receptors form multi-protein complexes for signaling integration (*Klingenberg, 1981*; *Salter, 2003*; *Altier et al., 2006*; *González-Maeso et al., 2008*). Among them, G protein-coupled receptors (GPCRs) are the most abundant and constitute the main targets in drug development (*Lagerström and Schiöth, 2008*). Although most GPCRs can signal in a monomeric form (*El Moustaine et al., 2012*; *White et al., 2007*; *Whorton et al., 2007*), increasing studies revealed that they can associate into both homo and heteromeric complexes (*Prezeau et al., 2010*; *Ferré et al., 2014*; *Gomes et al., 2016*; *Maurel et al., 2008*; *Albizu et al., 2010*). Interestingly, GPCR heteromers may generate original functional pharmacological entities different from each of the homomers (*Ferré et al., 2014*; *Gomes et al., 2016*; *Bellot et al., 2015*; *Wertman and Dupré, 2013*; *Fribourg et al., 2011*; *Urizar et al., 2011*). Whether such GPCR association is real in native tissue is still a matter of intense debate (*Bouvier and Hébert, 2014*; *Lambert and Javitch, 2014*).

Understanding how a receptor can control the activity of its partner (*González-Maeso et al., 2008*; *Albizu et al., 2010*; *Fribourg et al., 2011*; *Vilardaga et al., 2008*; *Han et al., 2009*) will certainly help clarify this important issue, opening ways to control the function of GPCR heteromers.

Metabotropic glutamate receptors (mGluRs) are class C GPCRs and are well-recognized constitutive homodimers (*Kniazeff et al., 2011*; *Pin and Bettler, 2016*). These receptors are divided into three groups: group I (mGlu$_{1,5}$), II (mGlu$_{2,3}$) and III (mGlu$_{4,6,7,8}$), on the basis of sequence homology, pharmacological profile and cellular signaling. Recently, mGluRs were shown to also form heterodimers with specific subunit composition (*Doumazane et al., 2011*; *Kammermeier, 2012*). Of note, whereas the different mGluRs were commonly described as having specific brain distribution supporting their homodimeric nature, localization studies revealed subcellular co-localization of different mGluRs such as mGlu1 and 5 (*Pandya et al., 2016*), mGlu2 and 4 (*Yin et al., 2014*), mGlu7 and 8 (*Ferraguti et al., 2005*). Further studies supported the existence of mGlu2-4 and 1–5 heterodimers in the brain (*Pandya et al., 2016*; *Yin et al., 2014*; *Moreno Delgado et al., 2017*). Being identified very recently, not much is known about the possible clinical relevance of mGlu heterodimers, but already homodimeric mGlu4, rather than heterodimeric mGlu2-4, were proposed as a better target for Parkinson's disease treatment (*Niswender et al., 2016*). In contrast, mGlu2-4 heterodimers control synaptic activity at the level of the cortico-striatal terminals in the striatum (*Yin et al., 2014*) and lateral perforant path terminals in the dendate gyrus (*Moreno Delgado et al., 2017*).

The mGlu subunits are multidomain proteins composed of a Venus flytrap domain (VFT) containing the orthosteric binding site, connected via a cysteine-rich domain (CRD) to a heptahelical domain (HD) involved in G protein coupling (*Pin and Bettler, 2016*; *Wu et al., 2014*; *Doré et al., 2014*; *Kunishima et al., 2000*; *Tsuchiya et al., 2002*). In the case of mGlu homodimers, structural and biophysical studies revealed a symmetrical conformational change during activation at the level of the VFTs (*Doumazane et al., 2013*; *Rondard et al., 2006*; *Huang et al., 2011*; *Xue et al., 2015*; *Olofson et al., 2014*; *Vafabakhsh et al., 2015*). Indeed, while activating one VFT is sufficient to partially activate the homodimers, activating both VFTs is required for full activity both in homodimeric (*Kniazeff et al., 2004*; *Brock et al., 2007*; *Levitz et al., 2016*) and heterodimeric receptors (*Moreno Delgado et al., 2017*). Surprisingly, this symmetric activation of the VFT dimer leads to an asymmetric activation of the HD dimer, only one HD being active at a time (*Hlavackova et al., 2012*; *Hlavackova et al., 2005*; *Goudet et al., 2005*). Whether mGlu heterodimer activation is symmetric or asymmetric, and whether either subunit can be involved in signaling remains unknown. Such analysis will likely bring interesting observation for the understanding of the allosteric coupling between GPCRs within hetero-complexes.

In this study, we choose mGlu2-4 heterodimer as a prototype heterodimer, as its existence in the brain has been documented (*Yin et al., 2014*; *Moreno Delgado et al., 2017*). We show that whereas both mGlu2 and mGlu4 HDs are capable of activating G proteins, only mGlu4 HD does it in mGlu2-4 heterodimers. Although a conformational change in the mGlu2 HD occurs that can be prevented by a mGlu2 negative allosteric modulator (NAM), it is not sufficient for a direct activation of G proteins, but important for the G protein coupling by the associated mGlu4 HD. This further documents the asymmetric activation of dimeric GPCRs. Furthermore, we demonstrated that manipulating the conformation of either mGlu2 or mGlu4 HD with positive and negative allosteric modulators can reorient the asymmetry towards mGlu2 activating G proteins. This demonstrates a differential ability of mGlu2 and mGlu4 HD to reach a G protein activating state. But most importantly, these data reveal strong allosteric interactions between two GPCRs in a dimeric complex. Such allosteric coupling can be controlled with small molecules allosteric modulators revealing a way to modulate heteromeric receptor activity, and expanding the possibilities of using such small molecules to precisely control signaling events. This illustrates how such hetero-complexes can control signals originating from various GPCR ligands targeting a cell.

## Results

### mGlu2-4 heterodimer activates G protein through mGlu4

The difficulty in studying GPCR heterodimers is that the co-expression of two different receptors leads to three populations of dimers, both homodimers and the heterodimer, making difficult the study of the specific properties of the heterodimeric entity. We then used a quality control system

that allows cell surface targeting of the heterodimer only (*Brock et al., 2007*). We engineered chimeric mGlu2 and mGlu4 subunits containing complementary coiled-coil regions (C1 and C2) derived from the GABA$_B$ receptor and intracellular retention signals (KKXX) (*Huang et al., 2011*; *Kniazeff et al., 2004*). We constructed mGlu2 and mGlu4 mutants with either C1 or C2, such as $^{HA}$mGlu2$_{C1KKXX}$ (2$_{C1}$), $^{Flag}$mGlu2$_{C2KKXX}$ (2$_{C2}$), $^{HA}$mGlu4$_{C1KKXX}$ (4$_{C1}$) and $^{Flag}$mGlu4$_{C2KKXX}$ (4$_{C2}$). By combination of C1 and C2 containing subunits, we can obtain the mGlu2-4 heterodimers specifically at the cell surface, as well as the mGlu2 (2-2) and mGlu4 (4-4) homodimers as controls (*Figure 1A,B*).

Such receptor constructs retained their ability to activate G proteins (we used the chimeric Gqi and Gqo proteins that enable coupling of these receptors to the PLC pathway [*Conklin et al., 1993*; *Blahos et al., 1998*]) when at the cell surface, with the expected action of mGlu2 (DCG-IV) and mGlu4 (L-AP4) selective agonists (*Figure 1—figure supplement 1*). When activated with glutamate, mGlu2 and mGlu2-4 displayed a similar potency and efficacy, while glutamate had a lower efficacy at mGlu4 (*Figure 1C*, *Figure 1—source data 1*). Note that all these receptor combinations were expressed at a similar level at the cell surface (*Figure 1B*).

Mutating the conserved Phe residue into Ser in the third intracellular loop of mGluRs abolished their ability to activate G proteins (*Kniazeff et al., 2004*; *Hlavackova et al., 2005*). When introduced into the C1 and C2 constructs, neither mGlu2-F756S, nor mGlu4-F781S activated G proteins (*Figure 1D,E*). When only one subunit carries the mutation, a larger decrease in glutamate efficacy was observed in mGlu2 dimers than in mGlu4 dimers (*Figure 1D,E*, *Figure 1—source data 1*). When such mutation was introduced in the mGlu2 subunit of the mGlu2-4 heterodimer, a glutamate-mediated response similar to the control was observed (*Figure 1D,F*, *Figure 1—source data 1*). In contrast, in the heterodimer containing the mutated mGlu4 subunit no response could be observed (*Figure 1F*) despite a correct expression level at the cell surface (*Figure 1—figure supplement 2*). As a control, introducing the FS mutation in both subunits of mGlu2-4 heterodimer abolished glutamate-induced G protein signaling (*Figure 1—figure supplement 3A*). Even when the selective mGlu2 and mGlu4 agonists DCG-IV and L-AP4 were used, signal could be generated with the heterodimers mutated in the mGlu2 subunit, but not in those mutated in the mGlu4 subunits (*Figure 1—figure supplement 3B–D*). Such data strongly suggest that in the heterodimer, G protein activation is exclusively mediated by mGlu4 HD. Of note, similar results were obtained with either mGlu2$_{C1}$-4$_{C2}$ or 4$_{C1}$-2$_{C2}$ heterodimers, indicating that the modified C terminal domains do not influence the asymmetric activation (*Figure 1C,F*).

## Asymmetric activation of mGlu2-4 HDs relies on the HDs only

In order to understand how a symmetric mGlu2-4 activation at the level of the VFT dimer could control an asymmetric activation of the HD dimer, we examined whether this could result from the specific association of one HD with its extracellular domain (ECD, composed of the VFT and the CRD). We then generated various constructs leading to the surface expression of receptor combinations composed of a 2–4 heterodimeric HD, but carrying either two mGlu2 ECDs (2-2$^{ECD}$4$^{HD}$; *Figure 2A*), two mGlu4 ECDs (4-4$^{ECD}$2$^{HD}$; *Figure 2B*), or in which the ECDs were swapped between the two subunits (2$^{ECD}$4$^{HD}$-4$^{ECD}$2$^{HD}$; *Figure 2C*). Constructs leading to the receptor combinations containing two mGlu2 VFTs (*Figure 2—figure supplement 1A*), two mGlu4 VFTs (*Figure 2—figure supplement 1B*), or in which the VFTs were swapped between the two subunits (*Figure 2—figure supplement 1C*) were also generated. For any of these combinations, we also analyzed the functional consequence of mutating either the mGlu2 or mGlu4 HD (*Figure 2* and *Figure 2—figure supplement 1*). Thanks to the C1 C2 terminal tails, a correct and specific expression of any of the indicated dimer combinations at the cell surface could be verified thanks to the HA or Flag N-terminal epitopes (*Figure 2—figure supplements 2* and *3*).

When activated by glutamate, we found that any combination carrying a wild-type mGlu4 HD (with or without a mutated mGlu2 HD) generated Ca$^{2+}$ signals (*Figure 2* and *Figure 2—figure supplement 1*). In contrast, none of those carrying a mutated mGlu4 HD were functional (*Figure 2* and *Figure 2—figure supplement 1*) despite their expression at the cell surface (*Figure 2—figure supplements 2* and *3*). Similarly, activating specifically the mGlu2 VFT with DCG-IV, or the mGlu4 VFT with L-AP4 led to G protein signaling of receptor combinations carrying at least one mGlu2 VFT, or one mGlu4 VFT, respectively, as long as the receptor contained a wild-type mGlu4 HD. No response could be generated with these agonists in receptor combinations carrying a mutated mGlu4 HD (*Figure 2* and *Figure 2—figure supplement 1*).

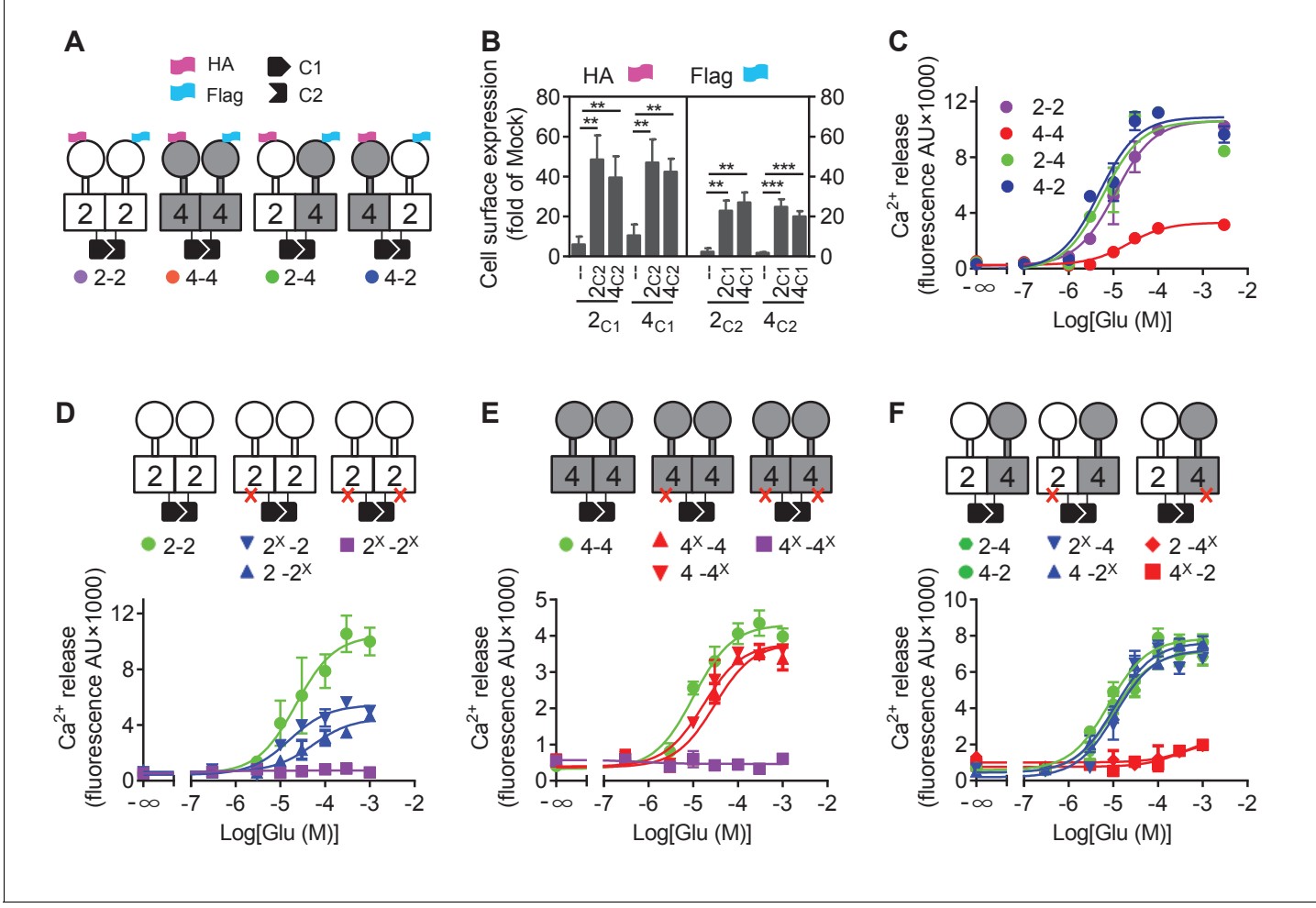

**Figure 1.** mGlu4 activates G protein in the mGlu2-4 heterodimer. (**A**) Cartoons illustrating mGlu2 and mGlu4 homodimers, and mGlu2-4 (2–4 or 4–2) heterodimers with each subunit carrying the quality control C1 or C2 system as C terminal tails, and the indicated HA or Flag tag at their N terminus. (**B**) Quantification of cell surface expressed HA-tagged or Flag-tagged constructs by ELISA on intact cells transfected with the indicated subunits ($2_{C1}$, $2_{C2}$, $4_{C1}$, $4_{C2}$) alone or together. Data are expressed as means ± SEM (n ≥ 3). **p<0.01, ***p<0.001 (unpaired t test). (**C, D, E, F**) Intracellular $Ca^{2+}$ responses mediated by the indicated subunits upon stimulation with increasing concentrations of glutamate, in the presence of the chimeric Gqi9, with the control subunits (**C**), the mGlu2 homodimer with no, one or both subunits mutated (**D**), same with mGlu4 homodimer (**E**) or mGlu2-4 heterodimer (**F**). The red cross indicates the subunit carries the FS mutation that prevents G protein activation. Data are expressed as means ± SEM of triplicates from a typical experiment repeated at least three times.

DOI: https://doi.org/10.7554/eLife.26985.002

The following source data and figure supplements are available for figure 1:

**Source data 1.** Glutamate potency at the indicated heterodimers.
DOI: https://doi.org/10.7554/eLife.26985.006

**Figure supplement 1.** G protein coupling of the mGlu subunits with C1 or C2 tail.
DOI: https://doi.org/10.7554/eLife.26985.003

**Figure supplement 2.** Cell surface and total expression of various mGlu dimer combination.
DOI: https://doi.org/10.7554/eLife.26985.004

**Figure supplement 3.** mGlu4 is responsible for G protein coupling in the mGlu2-4 heterodimer.
DOI: https://doi.org/10.7554/eLife.26985.005

We then examined the functional consequence of locking the ECD dimer in its active orientation. We previously reported that an inter-subunit disulfide bound between a Cys introduced at position 521 of the mGlu2 CRD led to a fully active dimer (mGlu2$^C$) (*Huang et al., 2011*). Mutating the equivalent position in mGlu4 (His523Cys) also generated a fully active receptor (mGlu4$^C$) (*Figure 3A*), as revealed by the accumulation of inositol monophosphate after LiCl addition. Note that the

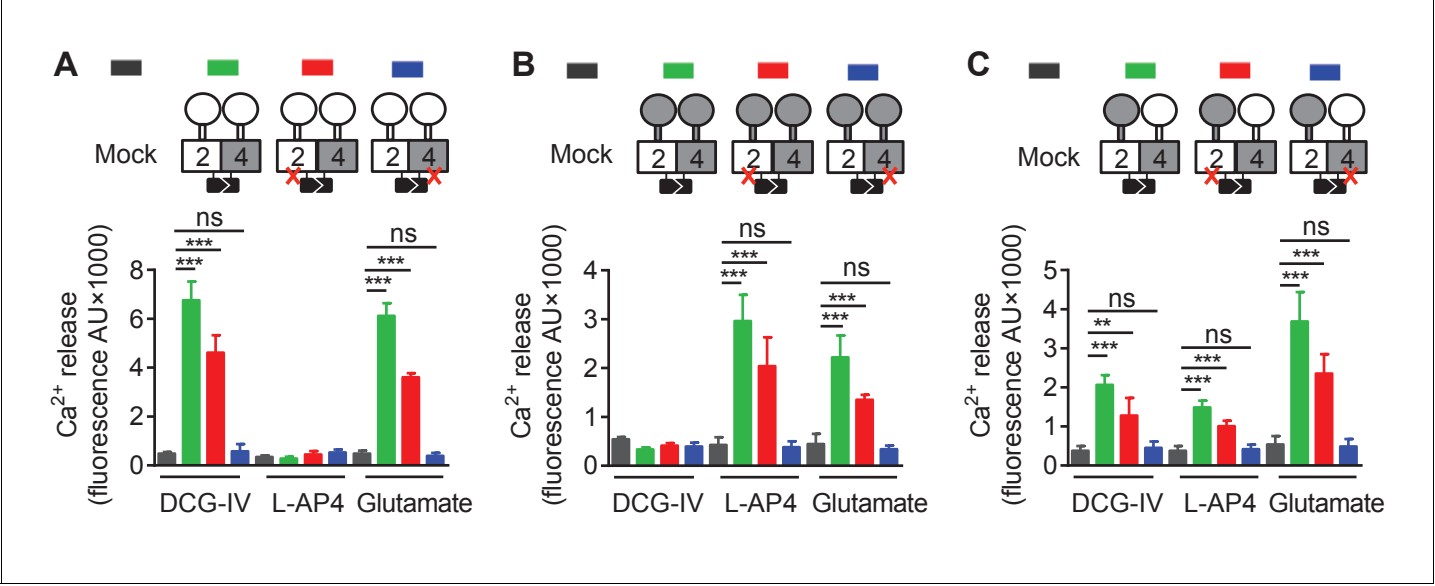

**Figure 2.** Asymmetric transduction results from the HDs in the mGlu2-4 heterodimer. In (**A**), (**B**) and (**C**) cartoons illustrating the heterodimer combinations used with one or both subunits carrying the FS mutation (red cross) that prevents G protein activation are indicated on the top. For each subunit, white domains are from mGlu2, while the grey domains are from mGlu4. The chimeric protein made of mGlu2 ECD and mGlu4 HD is named $2^{ECD}4^{HD}$ and the reverse chimera named $4^{ECD}2^{HD}$. The intracellular $Ca^{2+}$ responses mediated by the indicated subunit compositions (color coded, as indicated on top of the cartoons) upon stimulation with DCG-IV (30 µM), L-AP4 (30 µM) or glutamate (1 mM) shown at the bottom. (**A**) Data obtained with heterodimers containing both ECDs (VFT and CRD) from mGlu2. (**B**) Data obtained with heterodimers containing both ECDs from mGlu4. (**C**) Data obtained with heterodimers in which the ECDs were swapped between the two subunits. Data are means ±SEM (n ≥ 3). **p<0.01, ***p<0.001 (unpaired t test).

DOI: https://doi.org/10.7554/eLife.26985.007

The following figure supplements are available for figure 2:

**Figure supplement 1.** Asymmetric transduction results from the HDs in the mGlu2-4 heterodimer.
DOI: https://doi.org/10.7554/eLife.26985.008

**Figure supplement 2.** Cell surface and total expression of various mGlu dimer combination.
DOI: https://doi.org/10.7554/eLife.26985.009

**Figure supplement 3.** Cell surface and total expression of various mGlu dimer combination.
DOI: https://doi.org/10.7554/eLife.26985.010

constitutive activation of phospholipase C cannot be quantified through intracellular $Ca^{2+}$ measurements because $Ca^{2+}$ concentrations return close to basal values under constant PLC activity. When both mGlu2[C] and mGlu4[C] were co-expressed, an inter-subunit DTT-sensitive covalent linkage could be demonstrated, providing the natural inter-VFT disulfide bound is mutated (*Figure 3B*). Such a receptor combination also displayed a full constitutive activity (*Figure 3A*). In that case again, the high constitutive or glutamate-induced signaling could be observed in the heterodimer combinations containing a wild-type mGlu4 HD but not in those in which the mGlu4 HD contained the FS mutation (*Figure 3A*) despite a correct expression of all constructs (*Figure 3—figure supplement 1*)

Taken together, these results demonstrate that any ways the dimeric ECD of the mGlu2-4 heterodimer is activated the mGlu4 HD is always responsible for G protein activation. This strongly suggests that the asymmetric activity of the mGlu2-4 HD dimer is an intrinsic property of this membrane part of the receptor.

## mGlu2 HD is involved in the activation of mGlu4 HD

Although mGlu2 HD is not directly responsible for G protein activation in the mGlu2-4 heterodimer, it may still play a role in the activation process. We then used MNI137, an mGlu2 NAM (*Hemstapat et al., 2007*) known to stabilize the mGlu2 HD in its inactive conformation. We found that MNI137 partially inhibited mGlu2-4 heterodimer while it largely inhibited the response of the mGlu2 homodimer, whether the dimer was activated by glutamate (*Figure 4A*, *Figure 4—source*

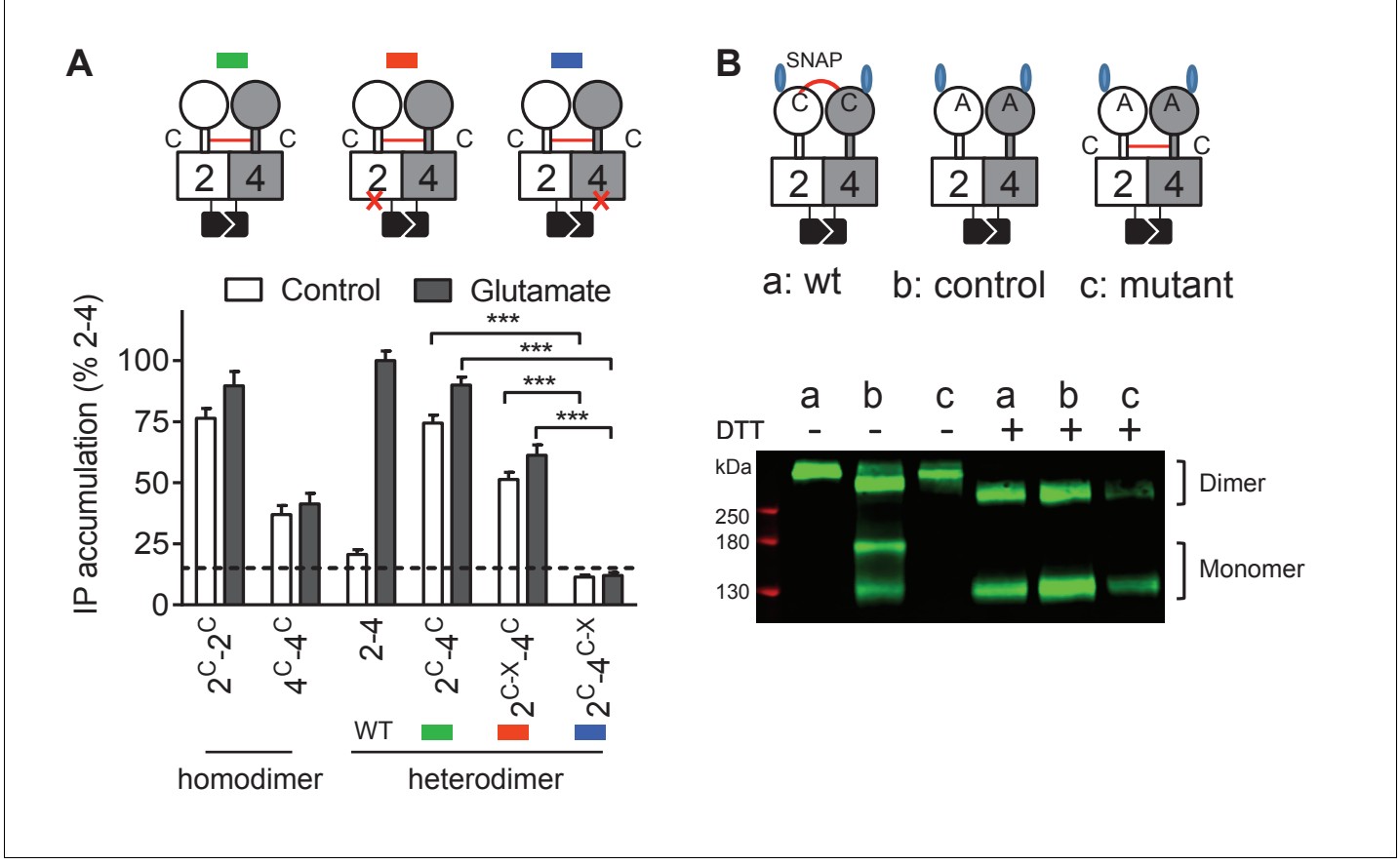

**Figure 3.** Constitutive activity of disulfide-tethered mGlu2-4 heterodimer is mediated by the mGlu4 subunit. (**A**) Cartoons illustrating the heterodimer combinations used with mGlu2 or mGlu4 subunits carrying the FS mutation (red cross) that prevents G protein activation (top). The red line linking both CRDs indicates the disulfide bridge that constrains the dimer into an active state. Inositol phosphate (IP) accumulation in cells expressing the dimer combinations after incubation with or without glutamate (1 mM). Data are means ±SEM (n ≥ 3). ***p<0.001 (unpaired t test). (**B**) On the top, the cartoons indicate the heterodimeric combinations analyzed by western blots (bottom) with or without DTT treatment. The natural inter-subunit disulfide bridge in the control dimer (wt) is indicated in (a), leading to the lack of monomers in the non-reducing conditions. When mutating both Cys involved in this natural crosslink (C121A in mGlu2 and C136A in mGlu4), both subunits can dissociate into monomers even in the absence of DTT (b). Adding a new disulfide bridge in the CRD (L521C in mGlu2 and H523C in mGlu4) (c) restores the subunit cross-linking. By using SNAP-tag labeling with a cell-impermeant fluorescein substrate, only the cell surface subunits are labeled, and then detected on the blot. Data are from a typical experiment repeated three times.

DOI: https://doi.org/10.7554/eLife.26985.011

The following figure supplement is available for figure 3:

**Figure supplement 1.** Cell surface and total expression of various mGlu dimer combination.

DOI: https://doi.org/10.7554/eLife.26985.012

*data 1*) or was constitutively active through the inter-CRD disulfide bridge (*Figure 4B*, *Figure 4—source data 2*). Of note, similar data were obtained in a receptor combination in which the mGlu2 subunit is incapable of G protein coupling. This revealed that a conformational change in the mGlu2 HD prevented by MNI137 binding is required to fully activate the mGlu4 HD.

To clarify how mGlu2 HD in the heterodimer controls mGlu4 HD activation, we activated the heterodimer with specific agonist of either mGlu2 or mGlu4. Interestingly, we found that MNI137 blocked mGlu2-4 heterodimer signaling induced by either DCG-IV or L-AP4 respectively whereas MNI137 inhibited only partially signaling induced by the combination of DCG IV and L-AP4 as observed with glutamate (*Figure 4A,C*, *Figure 4—source data 3*). These data revealed the prominent role of mGlu2 HD in activating mGlu4 HD when only one VFT is activated. Consistent with this conclusion, activation of both VFTs with DCG-IV in receptor combinations containing two mGlu2 VFTs is only partially inhibited by MNI137, with a smaller inhibition (39.2 ± 4.7%, n = 3) when both

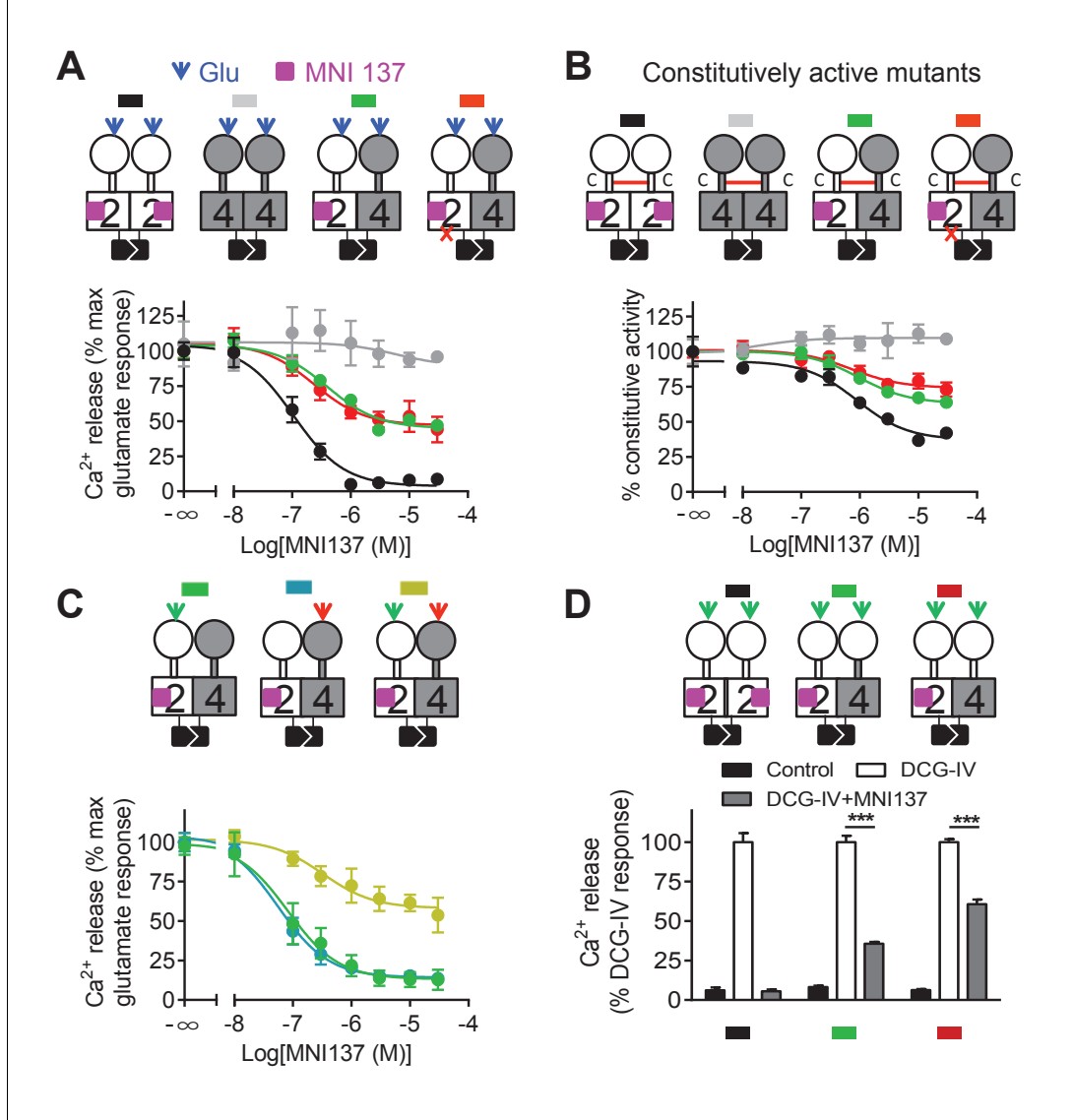

**Figure 4.** Allosteric regulation of mGlu4-induced signaling by mGlu2 HD. In each panel, cartoons (color coded) illustrating the dimer compositions used are indicated on the top, and intracellular $Ca^{2+}$ responses mediated by indicated dimer combinations upon stimulation with glutamate (1 mM) and increasing concentration of the mGlu2 NAM, MNI137 (purple square). The inactivating FS mutation is shown as a red cross. (**A**) Effect of MNI137 on homodimeric mGlu2 and mGlu4 receptors, and on the mGlu2-4 heterodimer carrying or not the FS mutation in the mGlu4 subunit activated by glutamate (blue arrow). (**B**) Effect of MNI137 on the constitutively active dimers resulting from the CRD disulfide cross-linking. (**C**) Effect of increasing concentrations of MNI137 on the mGlu2-4 heterodimer activated by the mGlu2 agonist DCG-IV (30 µM, green arrow), L-AP4 (30 µM, red arrow) or both. (**D**) Intracellular $Ca^{2+}$ response under control condition, or after stimulation with DCG-IV (30 µM, green arrow) with or without MNI137 (10 µM) with the indicated dimer combinations. Data are means ±SEM of triplicates from a typical experiment repeated at least three times (**A, B, C**), or from three independent experiments (**D**). ***p<0.001 (unpaired t test).
DOI: https://doi.org/10.7554/eLife.26985.013

The following source data is available for figure 4:

**Source data 1.** MNI137 potency at the indicated mGlu dimers.
DOI: https://doi.org/10.7554/eLife.26985.014
**Source data 2.** MNI137 potency at the indicated mGlu dimers.
DOI: https://doi.org/10.7554/eLife.26985.015
**Source data 3.** MNI137 potency at the indicated heterodimers.
DOI: https://doi.org/10.7554/eLife.26985.016

CRDs are from mGlu2, compared to the situation where the mGlu4 HD is associated with the mGlu4 CRD (64.3 ± 3.8% inhibition, n = 3, p<0.05) (*Figure 4D*).

## Allosteric control of the asymmetric activation of mGlu2-4 HDs

We then examined why mGlu2 HD could not mediate G protein activation within the mGlu2-4 heterodimer. We first analyzed the effect of an mGlu4 NAM (OptoGluNAM4.1) that we recently reported to prevent mGlu4 HD activation (*Rovira et al., 2016*). Surprisingly, this compound, while inhibiting agonist-mediated mGlu4 activity (*Rovira et al., 2016*), had no effect on the mGlu2-4 heterodimer (*Figure 5A*). Most interestingly, when using an mGlu2-4 combination in which the mGlu4 HD is unable to activate G protein, then a heterodimer unable to activate G proteins, the addition of OptoGluNAM4.1 allowed glutamate to generate a signal (*Figure 5A*, *Figure 5—source data 1*). This suggests that, by preventing mGlu4 HD to reach its active state, mGlu2 HD can take over for G protein activation in the heterodimer.

Such a proposal is supported by a second set of experiments, in which we favored mGlu2 HD activation in the mGlu2-4 heterodimer using the mGlu2 PAM LY487379. This compound had no

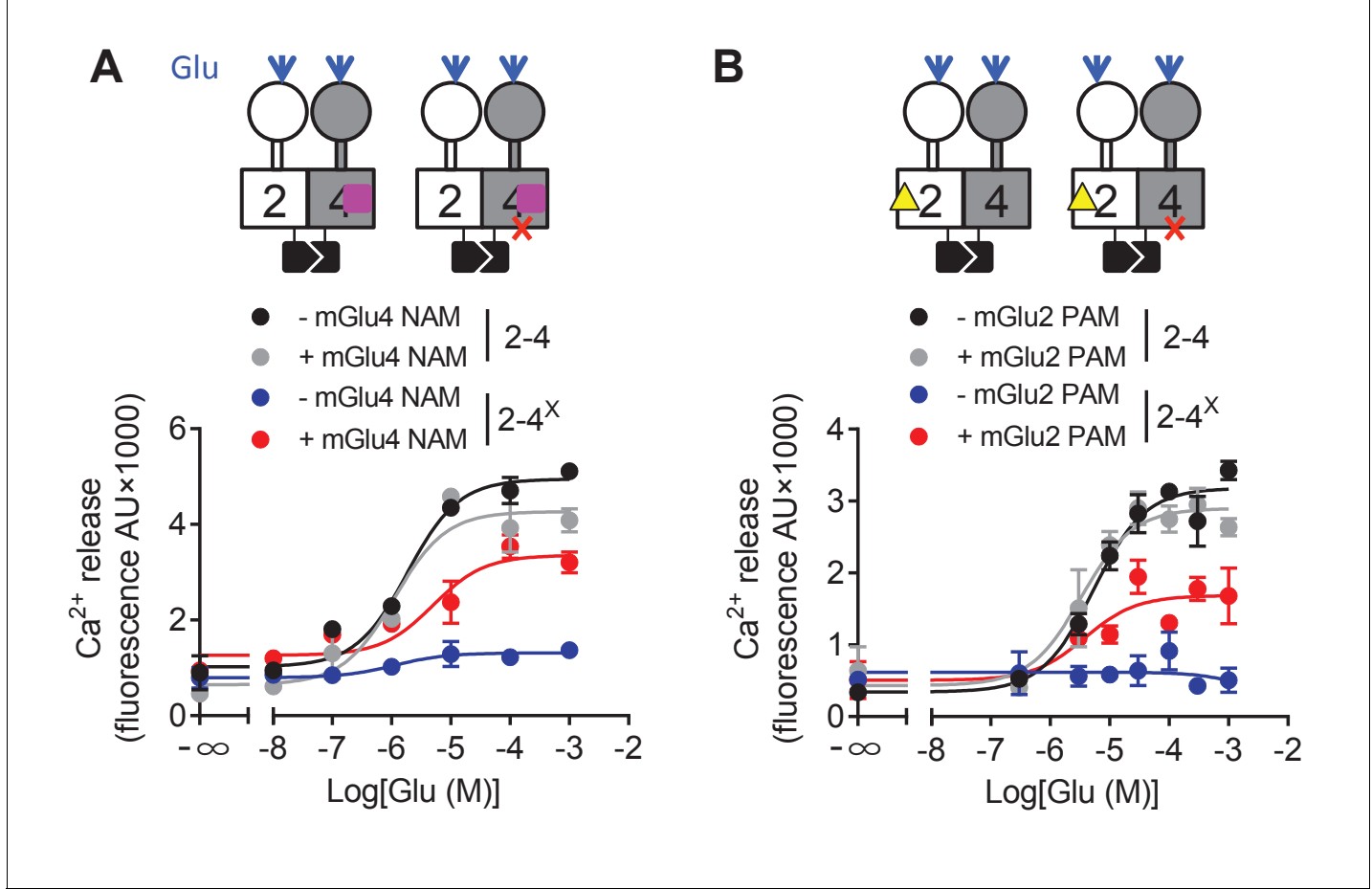

**Figure 5.** Switching of the G protein coupling subunit in the mGlu2-4 heterodimer by mGlu2 PAM and mGlu4 NAM. Intracellular Ca$^{2+}$ response mediated by the indicated subunits upon stimulation with increasing concentration of glutamate with/without a mGlu4 NAM (optoGluNAM4.1, purple square, 30 µM) or a mGlu2 PAM (LY487379, yellow triangle, 10 µM). (**A**) The mGlu4 NAM allows mGlu2 HD coupling to G proteins in the heterodimer. (**B**) The mGlu2 PAM allows mGlu2 HD coupling to G proteins. Data are means ±SEM of triplicates from a typical experiment repeated at least three times.

DOI: https://doi.org/10.7554/eLife.26985.017

The following source data is available for figure 5:

**Source data 1.** Glutamate potency at the indicated heterodimers and in the presence or absence of the indicated allosteric modulators.
DOI: https://doi.org/10.7554/eLife.26985.018

significant effect on the mGlu2-4 heterodimer signaling capacity (*Figure 5B*, *Figure 5—source data 1*) or on the mGlu2-4 heterodimer mutated in mGlu2 HD (*Figure 5—source data 1*). However, on the non-functional mGlu2-4 combination where the mGlu4 HD is mutated, the mGlu2 PAM LY487379 allowed glutamate to activate G proteins, therefore revealing a possible coupling of the mGlu2 HD in this heterodimer.

Taken together, these data revealed that under normal conditions, the coupling of mGlu2-4 is mediated by mGlu4 HD, but the mGlu2 HD can still generate signaling providing its activation is facilitated by a PAM, or by the inhibition of mGlu4 HD with a NAM (*Figure 5*).

## Asymmetric transduction of mGlu heterodimers

The above data were generated using mGlu2 and mGlu4 subunits carrying a modified C-terminal tail containing a quality control system. We then used another approach to validate our observation not only with mGlu2 and mGlu4 subunits with unmodified C-terminal tails, but also with other possible combinations of mGlu heterodimers (*Figure 6*). To that aim, we co-expressed two subunits, with one carrying the mutation preventing G protein activation. By specifically activating this mutated subunit, a functional response may only be generated with the heterodimer containing the wild-type subunit, providing the latter can be responsible for G protein activation in the heterodimer.

Using the non-functional mGlu4FS, co-expressed with mGlu2, no signal could be generated upon activation with the mGlu4 agonist L-AP4, in agreement with mGlu4 HD being the G protein-coupling domain in the mGlu2-4 heterodimer (*Figure 6A*). Consistent with this, activating an mGlu2FS mutant with DCG-IV generated a signal providing this subunit is co-expressed with mGlu4 (*Figure 6A*). Note that under these experimental conditions, cells expressed three types of dimers, the mGlu2 and mGlu4 homodimers and the mGlu2-4 heterodimer. Accordingly, if mGlu4 was the FS mutated subunit, L-AP4 had no effect, while DCG-IV could generate a signal through mGlu2 homodimers. In contrast, if mGlu2FS mutant was used, DCG-IV could generate a signal through the 2–4 heterodimer, and L-AP4 through the mGlu4 homodimers. Correct expression and function of all constructs was verified (*Figure 6—figure supplement 1*). These data confirmed that the asymmetric activation of the mGlu2-4 HDs is not the consequence of the presence of the C1-C2 intracellular domains.

The same approach was conducted with all possible heterodimeric receptors made of mGlu2 or mGlu3 associated with any of the group-III mGluRs: mGlu4, 6, 7 and 8. As depicted in *Figure 6C–F* and *Figure 6—figure supplement 2*, similar data were obtained with any of these 4 types of heterodimers for mGlu2 and for mGlu3. These data indicate that in all these cases, the group-III subunit is responsible for G protein activation in these heterodimers. As observed with the mGlu2-4 heterodimer (*Figure 4*), the mGlu2 NAM, MNI137, inhibited signaling of all these heterodimers (*Figure 6C–F*). This is consistent with the mGlu2 HD, though not directly involved in G protein coupling, being important to allow the group-III subunit to signal in these heterodimeric receptors.

## Discussion

Our data revealed important information on how two G protein-activating units communicate within a heterodimeric complex. We found that in the mGlu2-4 heterodimer, only the mGlu4 subunit activates G proteins, and we revealed a complex allosteric interaction between the two HDs. Indeed, the mGlu2 subunit retains its ability to signal providing its activation is favored using mGlu2 PAMs, or preventing the activation of mGlu4 HD with a NAM. Such findings, schematized in *Figure 7*, will certainly help elucidate the functional control of one GPCR by another, and how this can be modulated, then providing novel opportunities to decipher the role of possible GPCR heterodimers.

The key information reported here is that, even though both HDs in the mGlu2-4 heterodimer are capable of activating G proteins, only that of mGlu4 does it (*Figure 7*, State 2). This is not only observed with the mGlu2-4 but with any other heterodimers made of mGlu2 and a group-III mGlu subunits, where the group-III HD is always responsible for coupling. Such asymmetric activation of a GPCR dimer has often been observed (*Vilardaga et al., 2008*; *Han et al., 2009*; *Galvez et al., 2001*; *Duthey et al., 2002*; *Levoye et al., 2006*; *Xu et al., 2004*). Even in the mGlu homodimers, only one subunit is active at a time, although in that case, each subunit has the same probability of being active (*Hlavackova et al., 2012*; *Hlavackova et al., 2005*; *Goudet et al., 2005*). For the well-characterized heterodimeric GPCRs, such as the GABA_B and the T1R taste receptors, also one subunit only is responsible for G protein activation (*Galvez et al., 2001*; *Duthey et al., 2002*; *Xu et al.,*

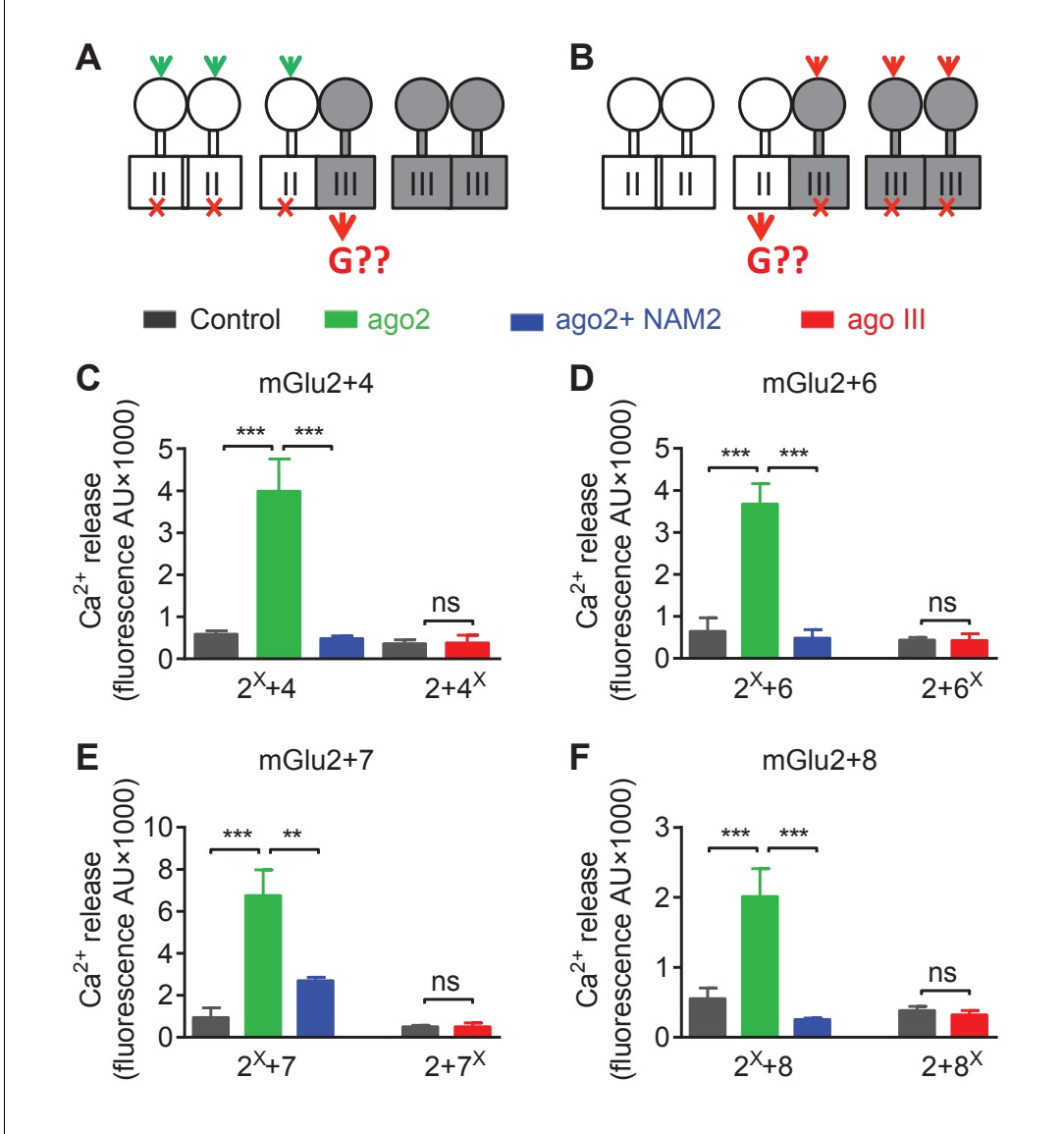

**Figure 6.** Asymmetric transduction by mGlu2-groupIII heterodimers. (A–B) schemes illustrating the method used to study the coupling properties of mGlu heterodimers composed of mGlu2 (group-II) and a group-III subunit with the wild-type C-terminal tails. In (A), activating specifically the mGlu2 subunit unable to activate G protein (F756S, red cross) can generate a signal only if associated with a functional group-III subunit. (B) Same as in (A) with the inactive group-III subunit (F781S, F773S, F784S, F777S in mGlu4-6-7-8, respectively) and a specific group-III agonist. (C–F) functional coupling of the indicated subunits under the condition indicated on the top (black, control; green, group-II agonist (DCG-IV, 30 µM); blue, group-II agonist with mGlu2 NAM (DCG-IV, 30 µM and MNI137 10 µM); red, group-III agonist (L-AP4, 30 µM for mGlu4-6-8, LSP4-2022, 300 µM for mGlu7). (C) Data obtained with cells expressing both mGlu2 and mGlu4, with either the inactive mGlu2 ($2^X$) or the inactive mGlu4 ($4^X$). (D, E and F), same as in C using mGlu6, mGlu7 or mGlu8 constructs, respectively. Data are means ±SEM (n ≥ 3). **p<0.01, ***p<0.001 (unpaired t test).

DOI: https://doi.org/10.7554/eLife.26985.019

The following figure supplements are available for figure 6:

**Figure supplement 1.** Expression and function of the indicated subunits.
DOI: https://doi.org/10.7554/eLife.26985.020
**Figure supplement 2.** Asymmetric transduction by mGlu2-groupIII heterodimers.
DOI: https://doi.org/10.7554/eLife.26985.021

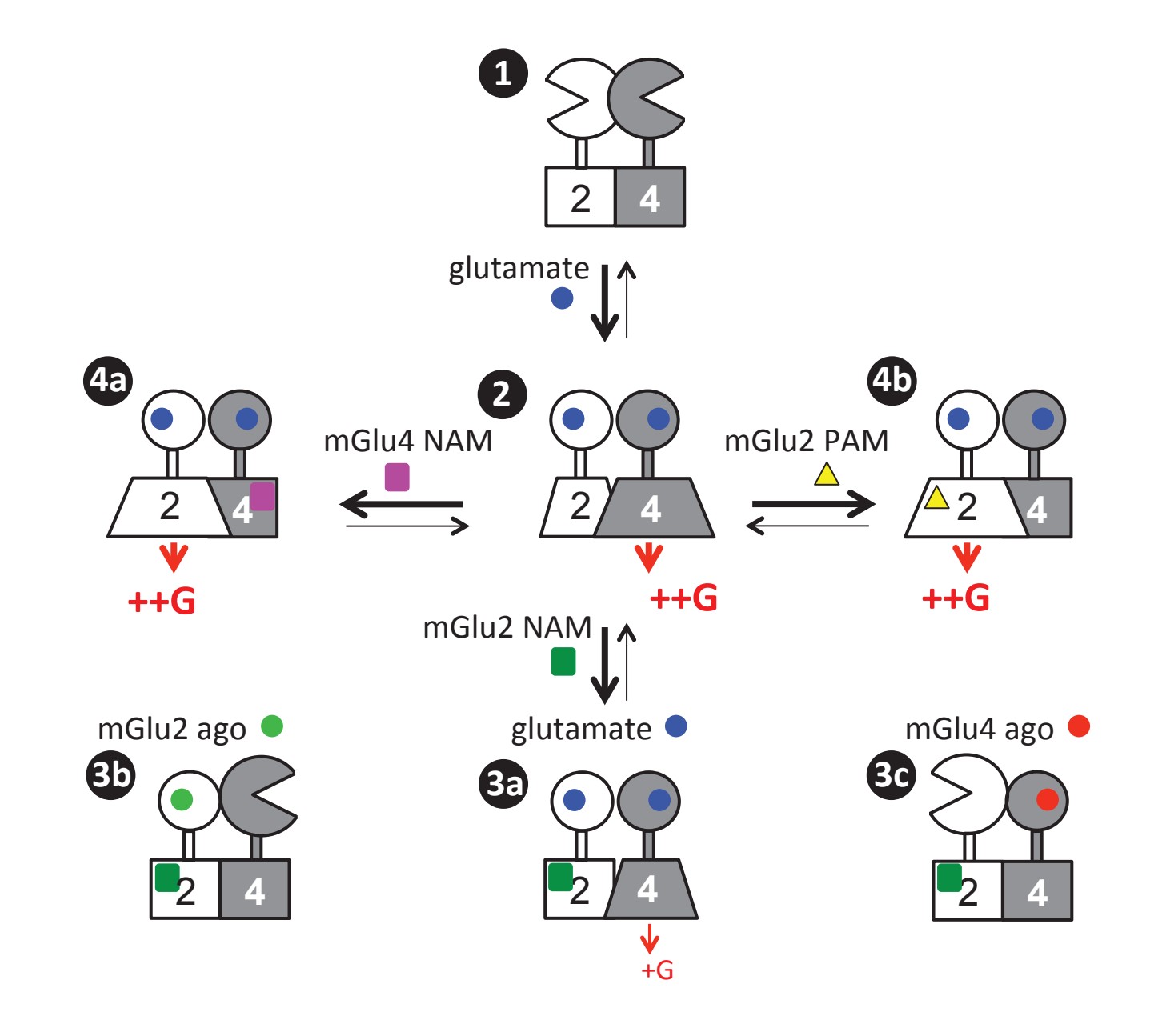

**Figure 7.** Scheme illustrating the activation mechanism and allosteric control of mGlu2-4 heterodimer. State 1. the inactive heterodimer in its basal state. State 2: Glutamate (blue disk) activation of both subunits leads to G protein activation by mGlu4 HD, also involving a conformational change in the mGlu2 HD. State 3: the addition of mGlu2 NAM (green square) largely decreases coupling efficacy of the mGlu2-4 heterodimer activated by glutamate (3a), or suppress detectable coupling if either the mGlu2 (3b) or mGlu4 (3c) is specifically activated. State 4: heterodimeric mGlu2-4 coupling through the mGlu2 HD thanks to the addition of a mGlu4 NAM (purple square, 4a), or a mGlu2 PAM (yellow triangle, 4b).

DOI: https://doi.org/10.7554/eLife.26985.022

2004), although this was assumed to result from the inability of the other subunit to signal. Our data suggest that instead, the inability of signaling of one subunit may result from an inhibitory effect of the other subunit while in its fully active state. This is indeed likely the case for the GABA$_B$ receptor for which the GABA$_{B1}$ subunit, not involved in coupling in the heterodimer, has been reported to signal when expressed alone (*Baloucoune et al., 2012*; *Richer et al., 2009*). Similar asymmetric coupling has also been reported for class A GPCR heterodimers, where the activation of one receptor

prevents the activation of the other (*Vilardaga et al., 2008*; *Han et al., 2009*; *Levoye et al., 2006*). These observations reinforce the idea of a strong negative cooperativity between the HDs in a dimeric GPCR complex, where the activation of one subunit suppresses the ability of the other to signal. This conclusion is further supported by numerous studies reporting negative cooperativity in agonist binding on dimeric GPCRs (*Albizu et al., 2010*; *Urizar et al., 2005*).

While mGlu4 homodimer coupling efficacy measured is lower than that of mGlu2, it is interesting to note that the coupling efficacy of mGlu2-4 heterodimer is similar to that of mGlu2 receptors. Because in both mGlu4 and mGlu2-4, the G protein activation is mediated by the mGlu4 HD, this means that the mGlu2 subunit potentiates mGlu4 efficacy. Such a low coupling efficacy of mGlu4 homodimers may well be the consequence of a weaker action of the mGlu4 ECD dimer on the HDs, consistent with the weaker coupling of heterodimers containing two mGlu4 ECDs (*Figure 2*). It may also possibly result in part from another level of interaction between the HDs within a dimeric receptor. Even though the mGlu2 HD does not directly activate the G protein in the heterodimer, it is important for the full activation of the mGlu4 HD (*Figure 7*). Indeed, the mGlu2 HD likely changes its conformation to exert this positive effect, even though this conformation is not sufficient for G protein coupling (*Figure 7*). This is well demonstrated by the partial inhibition of the mGlu2-4 activity (mediated by the mGlu4 HD) by a specific mGlu2 NAM known to stabilize the mGlu2 HD in its inactive state (*Hemstapat et al., 2007*) (*Figure 7*, State 3a). Such an action of the mGlu2 HD is even more prominent if the VFT dimer is asymmetrically activated, with only one VFT occupied by an agonist (*Figure 7*, States 3b, 3c). Such data revealed important allosteric interaction between two HDs in a GPCR dimer, not expected so far. They are however perfectly in line with the role of the GABA$_{B1}$ HD in the activation process of the heterodimeric GABA$_B$ receptor, the latter being involved in a direct activation of the GABA$_{B2}$ HD through an intra-molecular conformational change (*Monnier et al., 2011*).

Our data then further strengthen the multiple allosteric interactions between class C GPCR domains in the activation process, with the activation of one HD within the dimer being controlled in two ways. The first one is by the reorientation of the VFTs with an efficient coupling when the reorientation is symmetric (both VFTs activated), and a less efficient coupling in case only one VFT is activated (*Moreno Delgado et al., 2017*; *Kniazeff et al., 2004*). Of note, the sequence of the CRDs that link the VFTs to the HDs also plays a role, as indicated here by the differential coupling efficacies of receptor combinations containing either the mGlu2 CRD or the mGlu4 CRD (*Figure 2*, *Figure 2—figure supplement 1*, *Figure 4D*). The second pathway comes from a conformational change in the associated HD likely through a direct interaction between the two HDs in the dimer, consistent with our previous data with the heterodimeric GABA$_B$ receptor (*Monnier et al., 2011*). The second component is higher when the first, VFT mediated, is weak, due either to the activation of only one VFT (*Figure 4*), or to specific combinations of the CRDs (*Figure 2*, *Figure 4D*).

Such allosteric interaction most likely results from a contact between the two HDs via an interface that can communicate information from one HD to the other. This can be achieved if the interface involves component of the HD that changes conformation depending on the state of the subunit. Recently, we reported that, even though both HDs in a class C GPCR contact each other through TM4 and 5 in the basal state, as also reported for many class A GPCR dimers (*Guo et al., 2005*; *Manglik et al., 2012*), we surprisingly identified TM6 as being involved in the dimer interface of the active receptor dimer (*Xue et al., 2015*). TM6 is the TM that moves the most during class A GPCR activation (*Rasmussen et al., 2011a*; *Rasmussen et al., 2011b*), also likely in class C GPCRs (*Pin and Bettler, 2016*) for which the allosteric coupling between both HDs in a GPCR dimer is of fundamental mechanistic importance. More work is obviously needed to analyze the TM rearrangement in class C GPCR activation.

Another major observation is that it is possible, using small molecules to reorient the G protein coupling from one subunit to the other – i.e. from the mGlu4 HD to the mGlu2 HD in the mGlu2-4 heterodimer (*Figure 7*, States 4a, 4b). Indeed, by either preventing the activation of the mGlu4 HD with an mGlu4 specific NAM (*Figure 7*, States 4b), or by stabilizing the mGlu2 HD in its active conformation with a specific mGlu2 PAM (*Figure 7*, States 4b), the G protein coupling is transferred from the mGlu4 to the mGlu2 subunit. This observation suggests that the mGlu4 HD is likely more prone to reach a G protein activating state than the mGlu2 HD, and the fully active form of mGlu4 HD prevents the mGlu2 HD from reaching a G protein activating state. Such observation opens

interesting possibilities to decipher the specific role of each subunit in this mGlu2-4 heterodimer, especially in specific brain area where it is expressed.

There is much interest in the development of allosteric modulators, since these are expected to have less side effects for several reasons (*Changeux and Christopoulos, 2016*; *Conn et al., 2014*; *Foster and Conn, 2017*; *May et al., 2007*). Such small molecules target a site that is under less pressure during evolution, such that it is possible to identify subtype selective molecules, in contrast to compound acting in the orthosteric binding site highly conserved between homologous receptors. In addition, PAMs do not constantly activate the receptor, then do not favor receptor desensitization and internalization. Moreover, because they enhance the action of endogenous ligands, they increase the response when and where needed for an improved physiological response. Here we reveal a novel property of such small molecules: their ability to control the asymmetric activation of a GPCR dimer.

Taken together, the present study illustrates the complex allosteric interaction occurring between two associated G protein-activating units, with both positive and negative interactions. Indeed, a conformational change in one subunit is needed for a full G protein activation by its associated subunit, although the interaction also prevents the first subunit from activating G proteins. Because more and more data are consistent with the existence of GPCR heteromers, our finding will certainly bring much interest in elucidating their possible roles in integrating signals targeting either subunit.

# Materials and methods

## Materials

L-glutamate was purchased from Sigma. DCG-IV, L-AP4, MNI137 and LY487379 were from Tocris Bioscience. LSP4-2022 was a provided by Dr. F. Acher (Paris, France). Glutamate-pyruvate transaminase (GPT) was purchased from Roche. Lipofectamine 2000 and Fluo-4-AM were from Life Technologies. SNAP-Green was from NEN Biolabs.

## Plasmids and transfection

The pRK5 plasmids encoding the HA-tagged wild-type mGluR2-3-4-6-7-8 from rat were described previously (*Huang et al., 2011*). The site-directed mutations in the pRK5 plasmid were generated using QuikChange mutagenesis protocol (Agilent Technologies). The sequence coding C1 (the 47-residue coiled-coil sequence of the C-terminal of GABA$_{B1}$), or C2 (the 49-residue coiled-coil region of GABA$_{B2}$), followed by the endoplasmic reticulum retention signal KKTN. $^{HA}$mGluR2$_{C1KKXX}$ (2$_{C1}$) and $^{Flag}$mGluR2$_{C2KKXX}$ (2$_{C2}$) (with and without a N-terminal SNAP tag) have been reported previously (*Xue et al., 2015*). Using the same strategy, the last 38 residues in mGluR4 C terminus (HA, flag and SNAP-tagged versions of mGlu4 were used) were replaced by C1KKXX or C2KKXX to obtained $^{HA}$mGluR4$_{C1KKXX}$ (4$_{C1}$), $^{Flag}$mGluR4$_{C2KKXX}$ (4$_{C2}$). The chimeras (2$^{VFT}$4$^{HD}$, 4$^{VFT}$2$^{HD}$) were obtained by introducing a Bgl II restriction site in both mGlu2 and mGlu4 subunits, Ala497Arg mutation in mGluR2 and same sense mutations at Arg517Ser518 in mGluR4 were induced to make the restriction site. Chimeras (2$^{ECD}$4$^{HD}$, 4$^{ECD}$2$^{HD}$) were obtained by exchanging the ECD domain before the Pro557 in mGluR2 and Pro577 in mGluR4.

HEK-293 cells (ATCC, CRL-1573, lot: 3449904) were cultured in DMEM supplemented with 10% FBS and transfected by electroporation as described elsewhere. Absence of mycoplasma was routinely checkedusing the MycoAlert Mycoplasma detection kit (LT07-318 (Lonza, Amboise, France), according to the manufacturer protocol. Ten millions cells were transfected with 2 µg of each plasmid of indicated and completed to a total amount of 10 µg with the plasmid encoding the pRK5 empty vector. To allow efficient coupling of the receptor to the phospholipase C pathway, cells were also transfected with the chimeric G protein Gαqi9 (1 µg) or Gαqo (1 µg), and the glutamate transporter EAAC1 (1 µg). For cell-surface expression and functional assays of indicated subunits, experiments were performed after incubation for 36 hr (12 hr at 37°C, 5% $CO_2$ and then 24 hr at 30°C, 5% $CO_2$).

## SNAP fluorescent-labeled blot experiments

Cells after electroporation, adherent HEK293 cells plated in 12-well plates were labeled with 300 nM SNAP-Green in culture medium at 37°C for 1 hr. Cells were lysed with lysis buffer (50 mM Tris-HCl,

pH 7.4, 150 mM NaCl, 1% Nonidet P-40, 0.5% sodium deoxycholate, 0.1%SDS and protease inhibitors) at 4°C for 1.5 hr. After centrifugation at 12,000 g for 30 min, supernatants were added with loading buffer (NuPAGE LDSsample buffer 4, Invitrogen) for 10 min. Electrophoresis was performed using precast NuPAGE Novex 8% Tris-acetate gels (Life Technologies) and blotted onto nitrocellulose membranes. Membranes were imaged on an Odyssey infrared scanner (LI-COR Biosciences, Lincoln, NE, USA) at 800 nm for SNAP-Green (*Xue et al., 2015*).

## Cell surface quantification by ELISA

Cell surface expression of the indicated subunits was detected by ELISA. HA- and Flag-tagged subunits were co-transfected into HEK293 cells seeded into 96-well microplates. Cell surface expression and total expression (treated with 0.05% triton) was detected with a monoclonal rat anti-HA antibody (3F10, Roche) or rat anti-Flag (F1804, Sigma) and a goat anti-rat second antibody coupled to HRP (Jackson Immunoresearch, West Grove, PA) as previously described (*Monnier et al., 2011*). Bound antibody was detected by chemoluminescence using SuperSignal substrate (Pierce) and a 2103 EnVision Multilabel Plate Reader (Perkin Elmer, Waltham, MA, USA).

## Intracellular calcium release and inositol phosphate measurement

Intracellular $Ca^{2+}$ release was measured as described (*Hlavackova et al., 2005*). In brief, cells were pre-incubated for 1 hr with the $Ca^{2+}$ -sensitive Fluo-4 acetoxymethyl ester (Invitrogen). The fluorescence signals (excitation at 485 nm and emission at 525 nm) were then measured for 60 s (Flex-Station, Molecular Devices). Agonist was added after the first 20 s. The $Ca^{2+}$ response is given as the agonist-stimulated fluorescence increase. Concentration response curves were fitted using Graph Pad Prism.

Inositol phosphate (IP) accumulation in HEK293 cells co-transfected with indicated subunits was measured after stimulation with agonist for 30 min in 96-well microplates as previously described (*Hlavackova et al., 2005*). After incubation in the presence of LiCl (10 mM, 30 min) and termination of the reaction with 0.1 M formic acid, the supernatant was recovered and purified by ion exchange chromatography using DOWEX resin. Radioactivity was measured using a Wallac1450 MicroBeta microplate liquid scintillation counter (Perkin Elmer, Waltham, MA, USA).

## Statistical analysis

Statistical analyses were performed on at least three individual data sets analyzed by Graphpad prism using unpaired t-tests.

## Acknowledgements

This work was supported by the National Natural Science Foundation of China (NSFC grant nos. 31100548 to SH, and 31420103909, 31711530146, 31511130131), the Program of Introducing Talents of Discipline to the Universities of the Ministry of Education (grant no. B08029), Natural Science Foundation of Hubei province (grant no. 2014CFA010) and the Mérieux Research Grants Program of Institut-Mérieux (to JL). JPP was supported by the Centre National de la Recherche Scientifique, the Institut National de la Recherche Médicale and by the Fondation pour la Recherche Médicale (grant no. DEQ20130326522), and CisBio bioassays. JG was supported by the Ministerio de Economía y Competitividad (ERA-NET NEURON PCIN-2013–018 C03-02 and SAF2014-58396-R).

## Additional information

### Funding

| Funder | Grant reference number | Author |
|---|---|---|
| National Natural Science Foundation of China | 31100548 | Siluo Huang |
| Natural Science Foundation of Hubei Province | 2014CFA010 | Jianfeng Liu |

| | | |
|---|---|---|
| National Natural Science Foundation of China | 31420103909 | Jianfeng Liu |
| The program of introducing talents of discipline to the University of the Ministry of Education of China | B08029 | Jianfeng Liu |
| Mérieux research grants program | | Jianfeng Liu |
| National Natural Science Foundation of China | 31711530146 | Jianfeng Liu |
| National Natural Science Foundation of China | 31511130131 | Jianfeng Liu |
| Centre National de la Recherche Scientifique | | Jean-Philippe Pin |
| Institut National de la Santé et de la Recherche Médicale | | Jean-Philippe Pin |
| Fondation pour la Recherche Médicale | DEQ20130326522 | Jean-Philippe Pin |

The funders had no role in study design, data collection and interpretation, or the decision to submit the work for publication.

## Author contributions

Junke Liu, Data curation, Formal analysis, Investigation, Perform experiments; Zongyong Zhang, Data curation, Formal analysis, Perform experiments; David Moreno-Delgado, Conceptualization, Data curation, Formal analysis; James AR Dalton, Philippe Rondard, Conceptualization, Data curation, Formal analysis, Methodology, Writing—original draft; Xavier Rovira, Resources, Data curation, Methodology; Ana Trapero, Conceptualization, Resources, Data curation, Investigation; Cyril Goudet, Resources, Formal analysis, Methodology; Amadeu Llebaria, Conceptualization, Resources, Formal analysis, Supervision, Methodology, Writing—original draft; Jesús Giraldo, Formal analysis, Supervision, Methodology, Writing—review and editing; Qilin Yuan, Perform experiments; Siluo Huang, Conceptualization, Funding acquisition, Investigation, Writing—original draft; Jianfeng Liu, Conceptualization, Formal analysis, Investigation, Supervision, Funding acquisition, Writing—original draft, Writing—review and editing; Jean-Philippe Pin, Conceptualization, Formal analysis, Supervision, Funding acquisition, Writing—original draft, writing-review and editing

## Author ORCIDs

James AR Dalton http://orcid.org/0000-0002-5279-4581
Xavier Rovira http://orcid.org/0000-0002-9764-9927
Cyril Goudet http://orcid.org/0000-0002-8255-3535
Amadeu Llebaria http://orcid.org/0000-0002-8200-4827
Jesús Giraldo http://orcid.org/0000-0001-7082-4695
Philippe Rondard http://orcid.org/0000-0003-1134-2738
Siluo Huang http://orcid.org/0000-0002-8052-4677
Jianfeng Liu http://orcid.org/0000-0002-0284-8377
Jean-Philippe Pin http://orcid.org/0000-0002-1423-345X

## Decision letter and Author response

Decision letter https://doi.org/10.7554/eLife.26985.024
Author response https://doi.org/10.7554/eLife.26985.025

# Additional files

## Supplementary files

• Transparent reporting form
DOI: https://doi.org/10.7554/eLife.26985.023

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
