## [Decision Letter]

Thank you for submitting your article "Allosteric control of an asymmetric transduction in a GPCR heterodimer" for consideration by *eLife*. Your article has been reviewed by three peer reviewers, one of whom is a member of our Board of Reviewing Editors and the evaluation has been overseen by Richard Aldrich as the Senior Editor. The following individual involved in review of your submission has agreed to reveal his identity: Dmitry Veprintsev (Reviewer #2).

The reviewers have discussed the reviews with one another and the Reviewing Editor has drafted this decision to help you prepare a revised submission.

Summary:

This manuscript reports an investigation of the allosteric control between subunits in a heterodimer of metabotropic glutamate receptor 2 and 4, using an elegant and ingenious quality control system that assures surface expression of only the intended dimers. The study reports asymmetric activation of dimeric metabotropic glutamate receptors, such that if a mGlu2 subunit is present, mGlu4 is responsible for the G protein activation in the heterodimer. By using subtype specific agonists, as well as swapping the extracellular domains between the subtypes, the authors were also able to show that it is possible to activate signaling by the heterodimer via activation of the mGlu2 extracellular domain. Using constitutively-active cross-linked extracellular domains the authors also demonstrate that no matter how the extracellular domains are activated, it is the mGlu4 that is responsible for signaling.

Essential revisions:

The reviewers are enthusiastic about the manuscript. However, they raise a number of concerns that must be adequately addressed before the paper can be accepted.

1) Two of the reviewers raise serious doubts about the interpretation of the molecular dynamics simulations. These concerns can be summarized as: (1) the fluctuations of loop regions in a homology model during a simulation could have many origins unrelated to the activation mechanism, and since appropriate methodological details were not provided [the sequence identity of the template; the final alignment and curations of the alignment; the loop lengths and templates; structural reliability measures for those segments; providing the model as supplementary material], it is not clear that the models are sufficiently reliable in these regions to avoid such artifacts; (2) even if the models were reliable, it is entirely possible that differences in fluctuations of ~1 Å would not be robust to repeated, parallel simulations (n>1); and (3) critically, the underlying assumption that asymmetric dynamics in a dimer should be detectable as fluctuations on the timescale of microseconds in the monomer is speculative. For these reasons, this section of the manuscript should be removed before resubmission, and the conclusions adjusted accordingly.

2) The overall message of the manuscript should be communicated more consistently throughout the manuscript. In particular, from the data and from the discussion it is clear that some conformational change, but maybe not the full activation of mGlu2, is required for the allosteric activation of the mGlu4. In other places (abstract, end of introduction) this effect is described too simplistically as 'negative cooperativity'.

In addition, the statement in the abstract: 'revealed a dynamics "winner-take-all" mechanism in mGlu heterodimers, providing new insight on the allosteric control between subunits in a GPCR dimer’ implies some kinetic aspects and time-resolution of the measurements while no kinetic measurements were presented. Rewording this statement could help avoid confusion.

3) Cell-surface expression levels are not always clearly reported. Statistical analysis of the expression levels would aid in interpretation of the results, specifically in Figure 1—figure supplement 2 and Figure 2—figure supplement 2 and Figure 2—figure supplement 3. Similarly, in Figure 1, cell surface expression is only statistically compared with mock and not among the different heteromers. In this case, the data look similar, but this does not seem to be the case for several of the supplemental figures (see below).

4) Figure 1, Figure 2, Figure 4 and Figure 5 show the same type of data but use different Y axes (Fluorescence units or fold change over Mock, or normalised to the highest value). This makes it difficult to compare the experiments. It would be better to use the same Y axis – whichever is the most informative.

5) Figure 1 – the signal amplitude for the 2-4 heterodimer seem to be inconsistent on these two panels while it is consistent on others.

6) Figure 4 – The data presented are inconsistent with the previously presented data (compounded by the use of a different normalization along Y) and with the statement that the signaling is mediated via mGlu4. In the presence of NAM and a deactivating mutations signaling was observed. Panel D: are the differences in Ca^2+^ release between the two different 2-4 heteromers statistically significantly different? And if so: what could be an explanation for this (it seems that the inhibition of Ca^2+^ release is less profound when CRD is also from mGlu2)?

7) A more in-depth discussion of the following results is required:a) While it is concluded that the allosteric control of signaling is mediated via the 7TM domains, it is also worth discussing the possibility that some allosteric control could be mediated via the 7TM-VFT1-VFT2-7TM path, and only via the direct 7TM-7TM path. Three must be bi-directional allosteric connections between the 7TM and the VFT domains, and the allosteric coupling between the individual VFT domain has also been shown by the authors themselves.

b) In Figure 1: glutamate has a lower efficacy at mGlu4-4 compared to 2-2, 2-4, 4-2. Is that because 2 positively affects 4? This matter is not discussed. The high efficacy (Ca^2+^ release) of 2-4 and 4-2 in Figure 1 is not seen in Figure 1.

c) In Figure 1: why does mutation of one of the two monomers in mGlu2-2 lead to a larger decrease in glutamate efficacy than in mGlu4 dimers? Does mGlu2-2 couple to 2 G-proteins to reach highest activity, in contrast to mGlu4-4, 4-2 or 2-4 for which coupling to 1 G-protein is sufficient? Again, the use of different y-axes complicates ready comparison.

d) In Figure 1—figure supplement 3 C: why are there differences between 2x-4 and 4-2x? Could this be due to differences in expression level?

e) In Figure 2: the receptors consisting of VFT and CRD derived from mGlu2 consistently show a higher Ca^2+^ release. This fact is not discussed.

f) In Figure 3, it is not clear why IP accumulation is monitored in this particular experiment as opposed to Ca^2+^ release in the others.

g) In Figure 3—figure supplement 1, there are differences in expression levels: do they correlate with efficacy?

[Editors' note: further revisions were requested prior to acceptance, as described below.]

Thank you for submitting your revised work entitled "Allosteric control of an asymmetric transduction in a G protein-coupled receptor heterodimer" for consideration by *eLife*. Your revisions and responses have been reviewed by a Reviewing Editor, and the evaluation has been overseen by a Senior Editor.

We are willing to accept the manuscript for publication, provided that the molecular simulations are excluded. Although the nominal accuracy of the models on which the simulations are based appears to be reasonable (50% identity, high coverage), thereby addressing one of our previous concerns, it remains questionable whether the conclusions extracted from these simulations are sufficiently robust and convincing. The fundamental claim that microsecond-scale fluctuations of these two intracellular loops relate to the activation of a ligand-bound dimer is speculation – particularly for an apo, monomeric receptor. Critically, this claim is also not convincingly supported by the simulation data provided, based on the information added to the revision. Only one MD trajectory was calculated for the mGlu2 model, and although the trajectories are 5 microseconds-long, analysis of RMSD versus time shows the models continue to drift away from the starting structure throughout the simulation; therefore, it is unclear whether the observed fluctuations reflect functional dynamics or inaccuracies of the models. Indeed, no molecular explanation is given as to why one protein is "more dynamic" than another, leaving open the strong possibility that this difference owes to the differences in the quality of the models or how the simulations were set up or conducted. On top of this, the simulations include an apo monomer, rather than a ligand-bound dimer whose actions are being monitored in the experiments. In summary, the molecular simulations are insufficient to support the hypothesis put forward, and the relevance of the observations is too tenuous. It is our opinion that the speculation of the underlying mechanism for activation of mGlu4 vs mGlu2 is equally plausible without the simulations.

– There is a typo in the revised abstract: "their role in receptor remains elusive" (insert signaling)

– There is no reference to rat mGlu2 being used in the abstract, as indicated in the response regarding the title.

– In the Discussion section: The phrase: "the consequence of the mGlu4 ECD dimer", is confusing and possibly incomplete, since it refers also to data relating to mGlu2.

---

## [Author Response]

*Essential revisions:*

*1) Two of the reviewers raise serious doubts about the interpretation of the molecular dynamics simulations. These concerns can be summarized as: (1) the fluctuations of loop regions in a homology model during a simulation could have many origins unrelated to the activation mechanism, and since appropriate methodological details were not provided [the sequence identity of the template; the final alignment and curations of the alignment; the loop lengths and templates; structural reliability measures for those segments; providing the model as supplementary material], it is not clear that the models are sufficiently reliable in these regions to avoid such artifacts; (2) even if the models were reliable, it is entirely possible that differences in fluctuations of ~1 Å would not be robust to repeated, parallel simulations (n>1); and (3) critically, the underlying assumption that asymmetric dynamics in a dimer should be detectable as fluctuations on the timescale of microseconds in the monomer is speculative. For these reasons, this section of the manuscript should be removed before resubmission, and the conclusions adjusted accordingly.*

We agree with the referees that molecular modeling and especially MD simulations on 3D models highly depend on the quality of the models, and on the conditions used to conduct the molecular dynamics. We agree also that such data remains theoretical and can thus be criticized, unless supported by experimental data. In addition, we are well aware that the dynamics studies were performed using isolated HDs, not with the heterodimer. As such only part of what is undergoing is being analyzed since our studies did not consider the influence of one HD partner on the dynamics of the other.

With this in mind, and despite these well-deserved criticisms, we still consider our modeling study brings an interesting idea (not a conclusion) on what is possibly going on, leading to the G protein activation by one subunit (mGlu4) and not the other. We then propose to keep these data (now transferred to Appendix 1), supported by additional information, but we have modified our text, both results and discussion, to take such criticism into account, and to help the reader interpret these data with a degree of caution.

We have added more data to illustrate how the 3D models were generated, and how they have been validated (seq alignment details, templates, loop lengths etc.). We can also include homology model coordinates for the reviewer(s) to inspect (as separate files). Of note, the mGlu HDs share a similar target-template sequence identity (about 50%), such that 3D models are relatively accurate according to the actual methods used to generate them, especially the helices and their assembly. Specifically, an extra analysis of secondary structure has been included, which shows both 3D models have high structural stability during simulations. Recently, we used a similar approach to study the mode of action of the allosteric modulator MPEP that led us to propose a model to explain why MPEP is a NAM on mGlu5, and a PAM on mGlu4 (Dalton et al., 2017).

Because of the maintained symmetry at the level of the VFT, we speculated that the asymmetry at the level of the HD dimer comes from specific properties of the HD domains. One possibility being that the mGlu4 HD can reach the active state more easily, then requiring less energy, and as a consequence prevented the mGlu2 HD from reaching a fully functional state. Then, our in silico analysis mainly focused on the ability of the HD core to spontaneously “open” on its intracellular side, as an isolated domain in a lipid bilayer. In this respect we measure both distances between key intracellular loops and between key TM helices (the latter is a new addition). We find these measures to be related, although the difference between loops is more marked than between helices (~7 Å compared to ~2.5 Å). We also found that the mGlu2 3D model is more stable conformationally in its initial inactive state than mGlu4. Because both models were generated using the same methodology with similar target-template seq IDs, and were tested under the same in silico conditions (over long time periods), such an observation is consistent with our proposal. Even with the limitations mentioned above, we think these data provide a way for the reader to better understand what is possibly going on. It may also enable more focused future studies, either by ourselves or by others.

*2) The overall message of the manuscript should be communicated more consistently throughout the manuscript. In particular, from the data and from the discussion it is clear that some conformational change, but maybe not the full activation of mGlu2, is required for the allosteric activation of the mGlu4. In other places (abstract, end of introduction) this effect is described too simplistically as 'negative cooperativity'.*

Our observation that mGlu2 HD undergoes a conformational change, not sufficient for G protein activation, but necessary for a full activity of the mGlu4 HD is now presented clearly at the end of the introduction and at the beginning of the discussion. We extended a little the abstract to also indicate this point: “Such asymmetric transduction goes through both a direct activation of mGlu4 heptahelical domain (HD) by the dimeric extracellular domain, and an allosteric activation by a partially-activated non-functional mGlu2 HD”.

*In addition, the statement in the abstract: 'revealed a dynamics "winner-take-all" mechanism in mGlu heterodimers, providing new insight on the allosteric control between subunits in a GPCR dimer' implies some kinetic aspects and time-resolution of the measurements while no kinetic measurements were presented. Rewording this statement could help avoid confusion.*

The referee is correct that this way of presenting our conclusion implies some kinetic aspects that we did not examine. The sentence has been modified as follow: “These data revealed an oriented asymmetry in mGlu heterodimers that can be controlled with allosteric modulators. They provide new insight on the allosteric interaction between subunits in a GPCR dimer.”

We also modified the “One sentence summary” from “In mGlu heterodimers, the subunit that reaches first the active state couples to G protein, and prevents the other from signaling “to “In mGlu heterodimers, an oriented asymmetrical activation revealed complex allosteric interaction between subunit”.

*3) Cell-surface expression levels are not always clearly reported. Statistical analysis of the expression levels would aid in interpretation of the results, specifically in Figure 1—figure supplement 2 and Figure 2—figure supplement 2 and Figure 2—figure supplement 3. Similarly, in Figure 1, cell surface expression is only statistically compared with mock and not among the different heteromers. In this case, the data look similar, but this does not seem to be the case for several of the supplemental figures (see below).*

The figures reporting the cell surface expression have been simplified (we removed the total expression determined after cell permeabilization since these data did not bring much information), and made clearer to read. The main aim of these measurements was to check whether non-functional receptor dimer combinations were indeed expressed at the surface, to be able to conclude these receptors are not capable of activating G protein upon agonist stimulation. However, due to a non-linear relationship between cell surface expression and Fluo4 fluorescence intensity, we did not attempt to use this information to determine precisely the coupling efficacy of any given receptor combination. To do so, we should be sure we are in a linear relationship for both the signaling response measured and the cell surface receptor generated signal. By experience, we should use the IP One assay, rather than the Fluo4 assay, and use SNAP-tag surface labeling rather that sandwich ELISA for the cell surface quantification. Accordingly, we did not attempt to determine whether the variations on cell surface expression could be significant.

*4) Figure 1, Figure 2, Figure 4 and Figure 5 show the same type of data but use different Y axes (Fluorescence units or fold change over Mock, or normalised to the highest value). This makes it difficult to compare the experiments. It would be better to use the same Y axis – whichever is the most informative.*

Data in most figures are now raw data, expressed as the fluo4 fluorescence signal amplitude, as determined with a flexstation (Figure 1, Figure 2, Figure 5 and Figure 6). We however kept the normalized representation (% of maximal Glutamate effect, or% of constitutive activity) in Figure 4 in order to make clear the differential effect of the mGlu2 NAM MNI137 on the different receptor dimer combinations.

*5) Figure 1 – the signal amplitude for the 2-4 heterodimer seem to be inconsistent on these two panels while it is consistent on others.*

We apologize for this, since the original panel Figure 1 corresponded to an experiment where a different transfection protocol was used (lipofectamine, instead of electroporation). We now replaced this panel with data obtained under the same conditions as in the other panels, and as reported in the Materials and methods section. Note however, that differences can still be observed that correspond to variations between individual experiments as expected because of the transient transfection procedure. Since we decided to show raw data, we present data corresponding to means of triplicates from a typical experiment. Each experiment was reproduced at least three times. The mean potencies and relative maximum calculated from n independent experiments are indicated in Figure 1—source data file 1.

*6) Figure 4 – The data presented are inconsistent with the previously presented data (compounded by the use of a different normalization along Y) and with the statement that the signaling is mediated via mGlu4. In the presence of NAM and a deactivating mutations signaling was observed.*

We apologize for the mistake in Figure 4 and B. This is obviously not the mGlu4 subunit that was made incapable of signaling, but as stated in the text, the mGlu2 subunit. Indeed, the referee is correct, when the mGlu4 subunit is mutated, no signal can be measured, and then, no inhibition by MNI137. The figure has been corrected.

Panel D: are the differences in Ca^2+^ release between the two different 2-4 heteromers statistically significantly different? And if so: what could be an explanation for this (it seems that the inhibition of Ca^2+^release is less profound when CRD is also from mGlu2)?

The referee is right, MNI137 induced a smaller inhibition (39.2 ± 4.7%, n=3) when both CRDs are from mGlu2, compared to the situation where the mGlu4 HD is associated with the mGlu4 CRD (64.3 ± 3.8% inhibition, n=3, p<0.05). If we accept the idea that they are two main allosteric pathways leading to the mGlu4 HD activation, one coming from the mGlu2 HD, the other from the VFT dimer, we can propose that the VFT dimer component (that insensitive to MNI137) is higher in the receptor combination containing both CRDs from mGlu2. In other words, this suggests a better activation of the HDs by the dimeric mGlu2 ECDs. The influence of the CRD type in receptor activation in now discussed in our revised manuscript in a new paragraph added in the discussion.

*7) A more in-depth discussion of the following results is required:a) While it is concluded that the allosteric control of signaling is mediated via the 7TM domains, it is also worth discussing the possibility that some allosteric control could be mediated via the 7TM-VFT1-VFT2-7TM path, and only via the direct 7TM-7TM path. Three must be bi-directional allosteric connections between the 7TM and the VFT domains, and the allosteric coupling between the individual VFT domain has also been shown by the authors themselves.*

We have added a paragraph in the discussion to make this point clearer.

*b) In Figure 1: glutamate has a lower efficacy at mGlu4-4 compared to 2-2, 2-4, 4-2. Is that because 2 positively affects 4? This matter is not discussed. The high efficacy (Ca^2+^ release) of 2-4 and 4-2 in Figure 1 is not seen in Figure 1.*

It is true that using this Gqi9-mediated signaling assay, the mGlu4 homodimer displays a lower efficacy compared to the mGlu2 homodimer and mGlu2-4 heterodimer. This is consistent with data presented in Figure 1—figure supplement 1, and Figure 6—figure supplement 1 (previously Figure 7—figure supplement 1). As indicated by the referee, the simplest explanation is that the mGlu2 subunit increases the G protein-coupling efficacy of the mGlu4 HD. Alternatively we think this may simply be a specific property of the group-III dimeric VFTs. Indeed, the dimers in which both ECDs are from mGlu4 display a lower efficacy (see Figure 2 and Figure 6—figure supplement 1). A comparison with the data in Figure 2—figure supplement 1 even highlights the dimeric mGlu4 CRDs as being important for the low coupling efficacy. More work is clearly needed to better understand this observation. This is now discussed in the revised manuscript at the beginning of the third paragraph of the discussion.

Regarding the apparent efficacy difference between Figure 1, see our answer to the main point 5 above.

*c) In Figure 1: why does mutation of one of the two monomers in mGlu2-2 lead to a larger decrease in glutamate efficacy than in mGlu4 dimers? Does mGlu2-2 couple to 2 G-proteins to reach highest activity, in contrast to mGlu4-4, 4-2 or 2-4 for which coupling to 1 G-protein is sufficient? Again, the use of different y-axes complicates ready comparison.*

The referee is correct, mutating one subunit in mGlu2 dimer decreased receptor efficacy to 50% of the WT (Figure 1—source data file 1) as already reported with mGlu1 (Hlavackova et al., 2005). Such a decrease in signaling is proposed to be the consequence that only one HD is active at a time, with an equal probability of each subunit to reach an active state (see also Goudet et al., 2005). As such, a receptor dimer in which the mutated HD is in an active conformation does not couple, then leading to a decrease in coupling efficacy by about 50%. According to this proposal, the mutated HD does not activate the G protein, but is still able to reach an active conformation. It is then interesting to note that a lower decrease in efficacy is observed with mGlu4 upon mutation of one subunit (Figure 1—source data file 1). We added a new column in the table Figure 1-source data file 1 where the means maximal responses are presented. More work will be needed to understand why, but we can propose that the loss of function mutation on mGlu4 ic3 not only prevents G protein activation, but also reduces the ability of the HD to reach the active state, such that in the case of a mGlu4 dimer with only one subunit mutated, the wild-type HD will be more keen to activate G protein. We think it is not necessary to describe such a speculative hypothesis in the discussion.

*d) In Figure 1—figure supplement 3: why are there differences between 2x-4 and 4-2x? Could this be due to differences in expression level?*

It is true that in the presented experiment, DCG-IV generated a larger response on the 2x_C1_-4_C2_ receptor, than the 2x_C2_-4_C1_ receptor when the Gqi coupling is being analyzed. Note that such a difference is found neither when both DCG-IV and L-AP4 are applied (suggesting there is no major difference in expression level), nor when coupling to Gqo is being analyzed. When analyzing the data from the two other experiments performed, no such difference was found.

*e) In Figure 2: the receptors consisting of VFT and CRD derived from mGlu2 consistently show a higher Ca^2+^ release. This fact is not discussed.*

The referee is correct, any receptor combination made of two mGlu4 VFTs display a lower Ca^2+^ response, this is also the case for any other group-III mGluRs. See our answer to main point 7(b) above.

*f) In Figure 3: it is not clear why IP accumulation is monitored in this particular experiment as opposed to Ca^2+^ release in the others.*

In this figure, we analyzed the coupling capacity of receptor dimer combinations that are made constitutively active through a disulfide bond linking the CRDs in an active position. Because Ca^2+^ signals are transient, and return close to basal values after few minutes of constant activation, the constitutive activity of Gq coupled receptors cannot be measured with Fluo4 fluorescence assay. We then turned to the quantification of IP1 that can start to accumulate due to the constitutive activity of the mutant receptor, as soon as we block the IP1 degrading enzyme with LiCl. The reason for the choice of this assay is now clearly explained in the Results section.

*g) In Figure 3—figure supplement 1: there are differences in expression levels: do they correlate with efficacy?*

The surface expression levels were primarily measured to check whether non-functional combinations were indeed expressed at a correct level. Unfortunately, the relationship between the expression at the surface and the Fluo4 signal is not linear, due to a rapid saturation of the Ca^2+^ transient. Then trying to correlate expression level with maximal response in the Ca^2+^ assay is rather complex, and I think, not needed for the main take home message of this manuscript.

[Editors' note: further revisions were requested prior to acceptance, as described below.]

*We are willing to accept the manuscript for publication, provided that the molecular simulations are excluded. Although the nominal accuracy of the models on which the simulations are based appears to be reasonable (50% identity, high coverage), thereby addressing one of our previous concerns, it remains questionable whether the conclusions extracted from these simulations are sufficiently robust and convincing. The fundamental claim that microsecond-scale fluctuations of these two intracellular loops relate to the activation of a ligand-bound dimer is speculation –particularly for an apo, monomeric receptor. Critically, this claim is also not convincingly supported by the simulation data provided, based on the information added to the revision. Only one MD trajectory was calculated for the mGlu2 model, and although the trajectories are 5 microseconds-long, analysis of RMSD versus time shows the models continue to drift away from the starting structure throughout the simulation; therefore, it is unclear whether the observed fluctuations reflect functional dynamics or inaccuracies of the models. Indeed, no molecular explanation is given as to why one protein is "more dynamic" than another, leaving open the strong possibility that this difference owes to the differences in the quality of the models or how the simulations were set up or conducted. On top of this, the simulations include an apo monomer, rather than a ligand-bound dimer whose actions are being monitored in the experiments. In summary, the molecular simulations are insufficient to support the hypothesis put forward, and the relevance of the observations is too tenuous. It is our opinion that the speculation of the underlying mechanism for activation of mGlu4 vs mGlu2 is equally plausible without the simulations.*

*– There is a typo in the revised abstract: "their role in receptor remains elusive" (insert signaling)*

*– There is no reference to rat mGlu2 being used in the abstract, as indicated in the response regarding the title.*

– In the Discussion section: The phrase: "the consequence of the mGlu4 ECD dimer", is confusing and possibly incomplete, since it refers also to data relating to mGlu2.

As requested, we removed any reference to the MD simulations presented in Appendix 1, and of course we also removed the Appendix 1 and the related figures and files of the 3D models. The first sentence of the abstract has been modified as requested. We also added in the abstract that we worked with the rat mGluR sequences. We modified the sentences in the Discussion section, as follow: "Such a low coupling efficacy of mGlu4 homodimers may well be the consequence of a weaker action of the mGlu4 ECD dimer on the HDs, consistent with the weaker coupling of heterodimers containing two mGlu4 ECDs (Figure 2)".